# Unpredictable disturbance and its effects on activity behavior and lifespan in *Drosophila melanogaster*

Anabel S. X. Byars and Nicole C. Riddle*

## ABSTRACT

Animals exhibit natural movement patterns that are important for their survival and reproduction. Human disturbance can alter these movement patterns. In many natural settings, assessing the long-term impact of these altered movement patterns is difficult. Laboratory studies with model organisms may provide additional insight into the long-term effects of altered movement behaviors, such as those caused by human disturbance. Here, we investigate how unpredictable disturbance impacts animal activity and lifespan of *Drosophila melanogaster*. We selected four strains from the *Drosophila* Genetics Reference Panel (DGRP) to assess animals with different baseline activity levels. The unpredictable disturbance treatment was simulated using the TreadWheel to administer four randomized 30-min disturbances daily. We found that both 5-day and 20-day disturbance treatments altered activity levels, but the response was dependent on sex, genotype, and age of the animals. While we detected altered animal activity in several groups, lifespan generally was not affected, with a few exceptions. Our results highlight the complexity involved in predicting individual responses to disturbance but suggest that long-term effects on lifespan are rare in response to short-term disturbance.

KEY WORDS: Locomotor activity, TreadWheel, Circadian rhythms, *Drosophila melanogaster*

## INTRODUCTION

Animals in natural populations exhibit a wide range of movement behaviors. For example, these behaviors include activity to search for food and preferred habitats (Aikens et al., 2020; Bogdanović et al., 2021; Creel et al., 2005; Evans et al., 2016; Gill et al., 2009; Johansson et al., 2022; Lang et al., 2006; Lubitz et al., 2023; Merkle et al., 2016; Stutchbury et al., 2009; Zhang et al., 2018). Some herbivores migrate with seasonal changes in plant growth to improve their access to nutritious vegetation (Aikens et al., 2020; Merkle et al., 2016; Zhang et al., 2018). Predators shift their activity behavior when hunting prey or to adjust to changes in temperature (Johansson et al., 2022; Lang et al., 2006; Lubitz et al., 2023). In turn, prey shift their activity patterns and timing to avoid or escape predators (Creel et al., 2005; Culp et al., 1991; Johansson et al., 2022; Miyatake et al., 2004; Palmer et al., 2017; Randler and Kalb, 2020; Richardson et al., 2018). Many bird species migrate long distances over the winter to warmer climates (Gill et al., 2009; Stutchbury et al., 2009), and brown bears exhibit changes in activity levels leading up to, during, and following hibernation months (Bogdanović et al., 2021; Evans et al., 2016). Mating season is also a key time when natural movement patterns change, as the search for a mating partner begins (Akaarir et al., 2023; Bogdanović et al., 2021; Csányi et al., 2022; Gill et al., 2009; Pépin et al., 2009). For example, Pépin and colleagues monitored adult red deer using GPS collars in Cévennes National Park, France, and found that walking activity hit a peak for males during mating season, while females showed less activity in this period (Pépin et al., 2009). Similarly, Akaarir and colleagues tracked pearly razorfish in the waters of Palma Bay, Spain, during mating season and found that male fish exhibited higher activity levels starting earlier in the day compared to females (Akaarir et al., 2023). These examples illustrate how ecological factors influence animal movement behaviors and demonstrate how strongly animal activity patterns rely on predictable seasonal changes.

Another factor that has widespread effects on animal movement is human disturbance. Humans have altered many natural landscapes, often in unpredictable ways. Animals in natural populations come into human contact frequently, and even animals living on protected conservation sites are not fully shielded (Semper-Pascual et al., 2023). This human influence can affect animal movement patterns. However, whether or not human disturbance is perceived as stressful varies, as does the type of response, which can include increases, decreases, and no change to animal activity dependent on the species, type of disturbance, proximity to people, and more (Doherty et al., 2021; Gallagher et al., 2021; van der Kolk et al., 2024; Lamb et al., 2025; Lamichhane et al., 2023; Li et al., 2022; Pirotta et al., 2014; Salvatori et al., 2023; Smith et al., 2017; Thompson et al., 2013). For example, studies monitoring harbor porpoises' response to acoustic disturbance sometimes reported that the porpoises did not leave disturbed areas permanently, while others found potential reductions in foraging activity (Gallagher et al., 2021; Pirotta et al., 2014; Thompson et al., 2013). Similarly, Doherty and colleagues reported diverse effects on animal movement after surveying 208 studies encompassing 167 species (Doherty et al., 2021). While most of these studies document increased movement, especially when animals were trying to avoid unpredictable human threats like recreation, hunting, and aircrafts, some animals showed decreased movement, for example, in response to restrictive barriers that physically changed their habitat (Doherty et al., 2021). Camera trap surveys showed that both omnivore and prey activity were more likely to overlap with human activity, whereas predators were less active in human areas (Lamichhane et al., 2023; Li et al., 2022; Smith et al., 2017). Similarly, Kolk and colleagues revealed that how accustomed an animal is to a disturbance can predict their response, as shore birds accustomed to aircrafts responded little to the planes, whereas birds that encountered planes infrequently typically fled (van der Kolk et al., 2024). These studies illustrate that predicting the impact of human disturbance on activity patterns is

Department of Biology, University of Alabama at Birmingham, Birmingham, AL 35294, USA.

*Author for correspondence (riddlenc@uab.edu)

complicated, but many animals increase their activity levels at least temporarily to avoid humans.

Evaluating the impacts of altered movement behavior in natural populations can be challenging, as it requires measuring not just the movement alterations but also the quantification of the long-term effects on the animal. While some studies address this question in natural populations (Lamb et al., 2025; Salvatori et al., 2023), laboratory studies might be able to provide additional insights. In laboratory studies, different types of treatments that disrupt animal movement patterns and different lengths or frequencies of disturbance can be modeled, and their impact on organismal health can be measured. In the laboratory, unpredictable or predictable stressors can be assessed. While not all results from laboratory settings are expected to translate to natural populations, lab-based disturbance research can inform ongoing work in natural populations and lead to a deeper understanding of the factors determining how unpredictable alterations to movement patterns impact diverse animals.

*Drosophila melanogaster* is a promising model for investigating the consequences of increased activity levels due to disturbance. Compared to rodent models, *D. melanogaster* is inexpensive to culture in large numbers (Hales et al., 2015), and a lifespan of approximately 90 days (Piper and Partridge, 2018) makes it possible to track the effects of disturbance treatment over the flies' entire life. The natural activity patterns of *D. melanogaster* are well understood due to their use in circadian rhythm studies (Tataroglu and Emery, 2014). Additionally, their frequent use in studies on sleep deprivation, stress, and exercise, means established protocols for altering their movement patterns exist. Several groups have disrupted sleep by alterations to the light/dark cycle (Boomgarden et al., 2019; Vaccaro et al., 2016) or by mechanical perturbance during times of natural rest (Melnattur et al., 2020; Potdar et al., 2018), leading to abnormal activity behaviors. Heat stress (Koh et al., 2006), oxidative stress (paraquat) (Soares et al., 2017), and diet manipulations (Ko et al., 2017; Krittika and Yadav, 2020) also change animal activity levels in *D. melanogaster*. For example, a study investigating how stressful life events lead to increased risk of depression, exposed male flies to unpredictable bouts of cold, heat, starvation, and sleep stress over 10 days and saw reduced mobility (Araujo et al., 2018). Devices like the TreadWheel (Mendez et al., 2016) and PowerTower (Piazza et al., 2009) have been employed to manipulate animal activity for testing exercise regimens. Given these tools, *D. melanogaster* provides a great opportunity to mimic disturbance and investigate how altering activity levels impact animals both short-term and long-term.

In this study, we investigated how early-life unpredictable disturbance treatments impact movement behavior and lifespan in *D. melanogaster*. Four genotypes were selected from the *Drosophila* Genetics Reference Panel (DGRP) (Huang et al., 2014; Mackay et al., 2012) to represent a range of baseline activity levels from low to high (Riddle, 2020; Watanabe and Riddle, 2021a, b). We mimicked unpredictable disturbance with the TreadWheel, a device that induces movement, administering four randomized 30-min intervals of rotation each day (Mendez et al., 2016). We compared undisturbed controls to animals experiencing either 5 or 20 days of unpredictable disturbance treatment on the TreadWheel. We found that disturbance treatments can impact total activity, but the response type – increased, decreased, or no alteration – varies based on sex, genotype, and age. In addition, we found that the effect of disturbance can depend on the time of day and natural circadian rhythms of the animals, with some time periods more affected than others. Interestingly, for most experimental groups, there was little to no effect of disturbance on survival and lifespan.

Together, our results suggest the effect of disturbance on movement is context-dependent, strongly influenced by sex, genotype, age, and time of day, but that lifespan typically is not affected.

## RESULTS

To better understand the impact of unpredictable disturbance on health and lifespan, we mimicked disturbance using the TreadWheel. The TreadWheel rotates fly vials, which exploits *Drosophila*'s negative geotaxis behavior and induces movement (Mendez et al., 2016). By administering four randomized 30-min rotation periods per day over either 5 days or 20 days, we modeled varied lengths of unpredictable disturbance events. The four disturbance periods could occur at any time during the day or night, ignoring the flies' natural circadian rhythm. To assess the response to this disturbance, we used animals of both sexes from four DGRP lines with diverse baseline activity (Huang et al., 2014; Mackay et al., 2012). Our experiment included two outcome measures: First, we investigated the effects of disturbance treatments on overall animal health by measuring activity levels after the treatment ended. Second, we measured lifespan to determine if the survival probability of the flies was impacted by the disturbance (Fig. 1).

### Activity is dependent on genotype and sex

For this study, we chose DGRP301, DGRP304, DGRP786, and DGRP852 to compare animals with diverse activity patterns (Huang et al., 2014; Mackay et al., 2012). We specifically chose these four lines because our lab has characterized them extensively, and they have been used previously in a study with a predictable treatment schedule. To determine if the previously reported activity patterns persisted in this study, we used activity data from the control groups of each genotype on day 26. This day included all the treatment and control groups for the first time. As reported previously, we found that sex strongly impacts the animals' activity level ($P$=9.0e-12, Kruskal–Wallis rank sum test). Female flies moved less than their male counterparts in three out of four DGRP lines ($P$<3.4e-06 except for DGRP304, Kruskal–Wallis rank sum test; Fig. 2). Genotype was also a highly significant determinant of daily activity ($P$=3.2e-09, Kruskal–Wallis rank sum test). Generally, DGRP304 and DGRP786 females and males exhibited lower average mean activity levels. DGRP304 females and males, on average, crossed the vial midline 0.38 times and 0.46 times per 5-min interval over 24 h, while DGRP786 females recorded 0.03 and males 1.69 beam crossings, respectively (Fig. 2). In contrast, DGRP852 and DGRP301 animals exhibited higher average mean activity levels, with DGRP852 females recording 0.70 beam crossings and 2.91 in males, while females and males of DGRP301 recorded 0.92 and 4.38 beam crossings, respectively (Fig. 2). Analysis of variance (ANOVA) suggested that the interaction between sex and genotype strongly impacts activity ($P$<2.2e-16, Table S1). Together, these data confirmed that, as previously reported, genotype and sex strongly impact fly activity levels.

### Unpredictable disturbance treatments impact total activity in a context-dependent manner

Next, we examined the impact of 5-day and 20-day unpredictable disturbance treatments on animal activity. We measured animal activity over 24 h at four ages: days 11, 26, 37, and 51 (Fig. 1B, Fig. 3). Looking across all datasets, we found that sex, genotype, and age all significantly impact daily fly activity ($P$<2.2e-16; Kruskal–Wallis rank sum test). However, we did not detect a treatment effect on fly activity when combining data from all groups ($P$=0.052, Kruskal–Wallis rank sum test). To gain further insight,

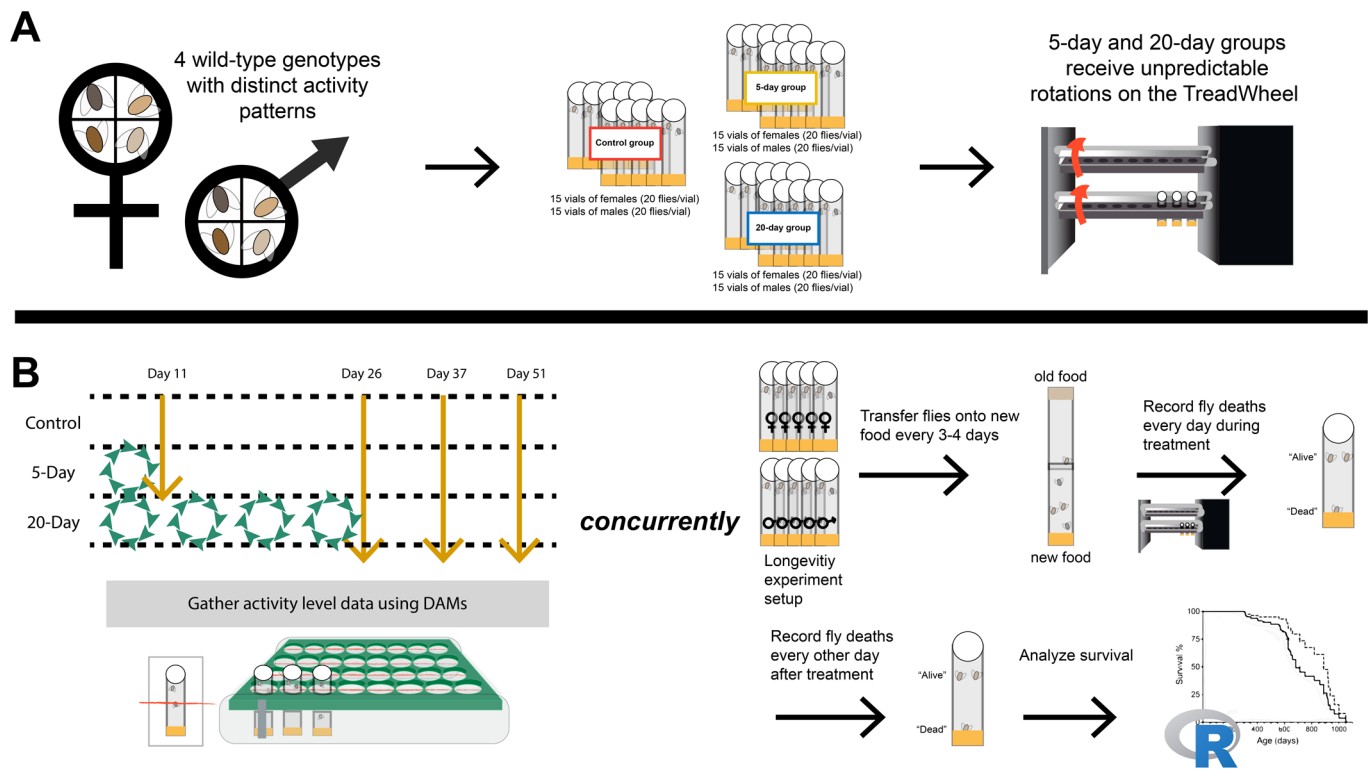

**Fig. 1. Experimental design.** (A) Four genotypes were used in this experiment, with each cohort consisted of control, 5-day, and 20-day treatment groups. For each group, there were 15 vials of unmated females and 15 vials of unmated males with 20 animals/vial. The 5-day and 20-day groups received 5 days and 20 days of unpredictable disturbance on the TreadWheel, respectively. Vials were randomly designated to control or disturbance treatment groups. (B) Activity measurements were taken on days 11, 26, 37, and 51. Flies were placed on the *Drosophila* Activity Monitoring (DAM) system immediately following the end of each disturbance treatment: day 11 and day 26. Then, flies were placed on the DAMs again on days 37 and 51. At the same time, fly deaths were counted every day during treatment and every other day after, until the last fly had died, to determine lifespan.

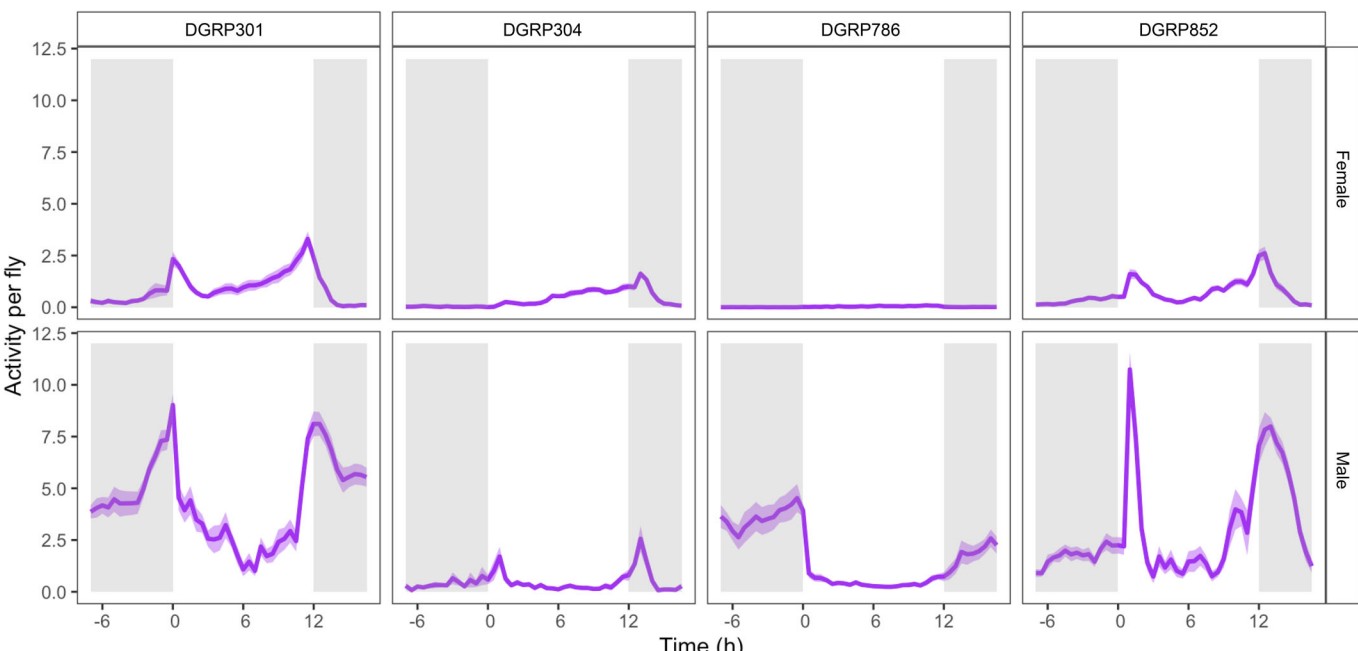

**Fig. 2. Activity is highly dependent on genotype and sex.** Activity per fly (y-axis) is shown for control animals aged 26 days from four genotypes: DGRP301 (*n*=31 vials), DGRP304 (*n*=27 vials), DGRP786 (*n*=32 vials), and DGRP852 (*n*=31 vials) plotted against time (x-axis) in hours. Activity is measured using DAMs, which record the number of times flies cross the midsection of a vial every 5 min over 24 h. The activity is normalized per fly. Data from females (top) and males (bottom) are shown separately. The dark purple line represents average activity, and the light purple shading represents standard error. The grey shaded areas indicate lights off, and the white areas indicate lights on, with time 0 marking lights on at 07:00.

we used ANOVA, which indicated that sex, genotype, treatment, and age impacted activity through interaction effects. Specifically, sex, genotype, and age showed a significant three-way interaction ($P$<2.2e-16; Table S2), while treatment affected activity in a sex-specific manner ($P$=0.00024; Table S2). The ages at which we saw significant disturbance treatment effects compared to controls were on days 11 and 51 for DGRP304 males, days 37 and 51 for DGRP304 females, day 37 for DGRP786 males, and day 37 for

DGRP852 females (Fig. 3). Clearly, sex, genotype, and age are important determinants of how disturbance influences activity levels.

### Unpredictable disturbance treatment effects differ between female and male flies with the age of the animals

We also wanted to understand how the response to disturbance changes as the animals age and are further removed from the disturbance event. Thus, we compared activity levels immediately

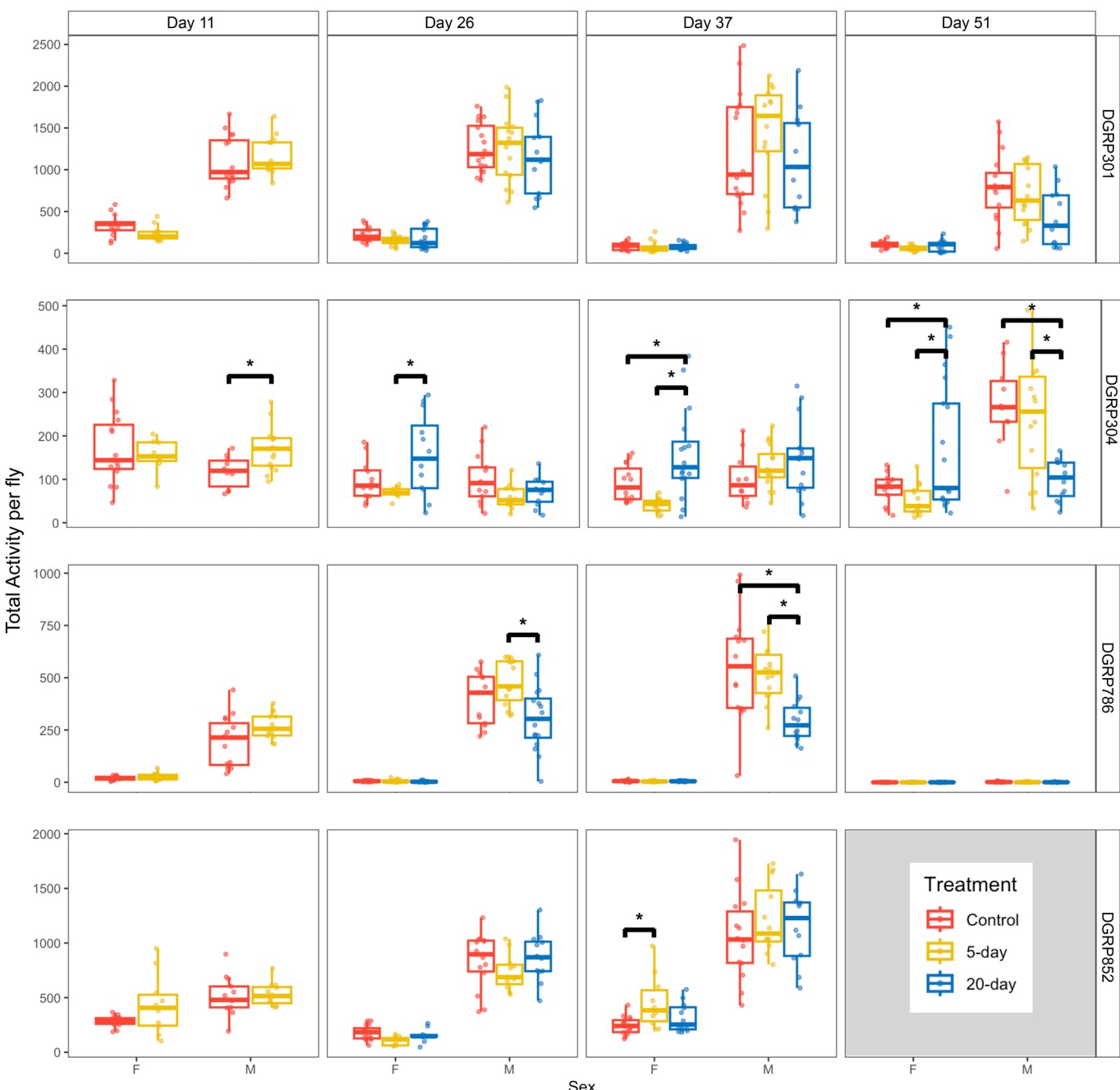

**Fig. 3. Unpredictable disturbance impacts total activity in a context-dependent manner.** Total activity per fly (y-axis) is plotted against sex (x-axis). Activity is measured using the DAMs, which record the number of times flies cross the beam at the midsection of the vial every 5 min. Then, activity is normalized per fly and totaled over 24 h. Data from the control are shown in red, 5-day disturbance treatment are shown in yellow, and data from the 20-day disturbance treatment are shown in blue. The four ages, day 11, 26, 37, and 51 are plotted across the top. The four genotypes, DGRP301 (*n*=29-31 vials/group), DGRP304 (*n*=27-32 vials/group), DGRP786 (*n*=31-32 vials/group), and DGRP852 (*n*=23-31 vials/group), are plotted along the right side. The box plots show the 25th and 75th percentile range with whiskers extending to include all data points up to 1.5x the standard deviation. The center line represents median activity. * indicate significant differences (*P*<0.05, pairwise *t*-test). Note that the y-axis is not scaled across genotypes.

following the 5-day disturbance treatment on day 11 and immediately following the 20-day disturbance treatment on day 26 with activity readings at two later time points on day 37 and day 51. For context, day 51 was approximately 'mid-life' for the flies used in this experiment. We analyzed the data for each genotype separately, again using ANOVA. For each genotype, we found that sex and age interact to affect activity ($P<8.11e-08$, $F$-test). Two of the genotypes, DGRP786 and DGRP304, also showed a three-way interaction between sex, age, and treatment on activity levels ($P<6.34e-05$, $F$-test), illustrating again that responses in activity to these disturbance treatments were highly context-dependent.

Though fly activity levels varied immensely from group to group, there were some common trends. For each genotype, females showed a gradual decrease in their total activity levels from day 11 to day 51 (Fig. 3). This pattern was observed in the control, 5-day, and 20-day groups. The lowest activity levels were seen for DGRP786 females, which showed a 97% reduction in mean activity in the 5-day disturbed flies, a 95% reduction in the 20-day disturbed flies, and a 96% reduction in the control flies from day 26 to day 51 (Fig. S1A). In contrast, the males often fluctuated between increased and decreased activity levels with age (Fig. 3). For example, DGRP852 males in all treatment groups showed an increase in mean activity levels up to day 37 (Fig. 3). Similarly, we saw an increase in mean activity levels for the DGRP304 males over time, with levels being ~4.1× higher in the 5-day disturbed group and ~2.7× higher in the control group on day 51 versus day 26 (Fig. 3). However, DGRP786 and DGRP301 males showed a different trend. Their 5-day disturbed groups had an increase in movement from day 26 to day 37, followed by an abrupt decrease in movement on day 51 (Fig. 3). Our results reveal that changes in activity level with age depend strongly on sex (Table S2).

### Unpredictable disturbance treatment can increase, decrease, or have no effect on total activity

Given that disturbance treatment overall had an impact on fly activity through interaction effects (Table S2), we sought to identify the specific treatment effects in the individual experimental groups (sex/genotype/age). We detected several significant treatment effects on total activity but again noticed that the effect varied based on sex, genotype, and age (Fig. 3). There was no significant effect of disturbance treatment among DGRP301 flies ($P<0.77$, Kruskal–Wallis rank sum test). For DGRP304, we saw immediate effects of disturbance among the males (Fig. 3), with a 45% increase in mean activity following the 5-day treatment compared to controls ($P=0.01$, pairwise $t$-test). Several weeks later, on day 51, activity levels for males were reduced greatly in the 20-day group compared to the 5-day group and controls ($P=0.0047$ and $P=0.0014$, pairwise $t$-test). For DGRP304 females, the response to 20-day unpredictable disturbance was opposite that of males. On days 26, 37, and 51, females exhibited major increases in activity compared to the 5-day group (days 37 and 51) and control (day 37) ($P=0.014$, $P=0.00012$, $P=0.0038$, and $P=0.028$, pairwise $t$-test), illustrating that sex was an important factor in the treatment response (Fig. 3). In DGRP786, activity levels of males were impacted by disturbance (Fig. 3), but there was no impact for females (Fig. S1A). On day 26 ($P=0.0022$, pairwise $t$-test) and day 37 ($P=0.0045$, pairwise $t$-test), comparing the 5-day to 20-day groups, DGRP786 males receiving 5-day disturbance treatment showed higher activity, while those receiving 20-day disturbance treatment showed lower activity. Additionally, mean activity for the 20-day group saw a 47% reduction compared to the controls on day 37 ($P=0.0012$, pairwise

$t$-test) (Fig. 3). For DGRP852 females, activity levels were altered by the disturbance treatment. Almost 4 weeks after the 5-day disturbance ended, DGRP852 females from the 5-day group moved 86% more than controls ($P=0.0054$, pairwise $t$-test; Fig. 3). These results demonstrated that there was no general pattern for how 5-day or 20-day unpredictable disturbance treatments will impact total activity levels. Instead, depending on sex, genotype, and age, unpredictable disturbance can increase, decrease, or have no impact on animal activity. Effects can occur immediately after the treatment or may be delayed. While some changes in activity levels persisted, others were transient.

### Unpredictable disturbance treatment effect on animal activity depends on time of day (07:00, 13:00, and 19:00)

In the analysis presented in Fig. 3, we investigated the impact of unpredictable disturbance treatments on total activity over a 24-h period. However, the activity levels of *D. melanogaster* have distinct circadian rhythms, showing increased activity when the light changes and low activity during the main light and dark phases. Thus, we asked if unpredictable disturbance treatment effects depend on the time of day. Specifically, we investigated the hours around 07:00 and 19:00, when the lights in the incubator turn on and off, and the flies typically show peak activity. For comparison, we also included a 2-h window around 13:00, which represents a time of low activity level. Thus, we analyzed activity data on day 26 from 06:00-08:00, 12:00-14:00, and 18:00-20:00 using ANOVA and pairwise $t$-tests to look for time-dependent treatment effects. First, we noticed that the effect of unpredictable disturbance was not always detectable at all three time periods, indicating that the treatments administered do not simply increase or decrease activity levels equally throughout the day. For example, when analyzing activity over a 24-h period, no significant treatment effects were detected for DGRP301 ($P=0.77$, Kruskal–Wallis rank sum test). However, we found that the 5-day group in DGRP301 females moved 42% less than controls, specifically during the 18:00-20:00 window ($P=0.027$, pairwise $t$-test; Fig. 4A). DGRP304 females from the 20-day treatment group moved significantly more than the 5-day treatment group across all three time periods ($P<0.014$, pairwise $t$-test). For DGRP304 females, activity levels steadily increased throughout the day. For example, the 20-day DGRP304 females, on average, crossed the vial midline 0.27 times at 06:00-08:00, 0.90 times at 12:00-14:00, and 1.74 times at 18:00-20:00 per 5-min interval. Only one significant treatment effect was detected among DGRP304 males from 18:00-20:00, with the 5-day group showing reduced activity levels compared to controls ($P=0.0089$, pairwise $t$-test; Fig. 4B). In DGRP786, males experiencing the 20-day disturbance treatment consistently had lower activity levels compared to the 5-day disturbance group at all three time periods ($P<0.026$, pairwise $t$-test) and compared to the controls at the 06:00-08:00 ($P=1.1e-05$, pairwise $t$-test) and 12:00-14:00 ($P=0.007$, pairwise $t$-tests) time periods (Fig. 4C). For the DGRP786 females, a significant treatment effect on activity levels was detected from 12:00-14:00, with again, the 20-day disturbance decreasing movement compared to controls ($P=0.026$, pairwise $t$-test; Fig. S1B). Both Fig. 4B and C demonstrate that if the effect of disturbance can be detected throughout the day, the direction of change – increase or decrease activity – is consistent across time points, even though baseline activity levels vary depending on the time of day.

In other treatment groups, animal activity was altered at specific time points. For example, the average activity of DGRP852 females was ~35% lower in the 5-day treatment group at 12:00-14:00

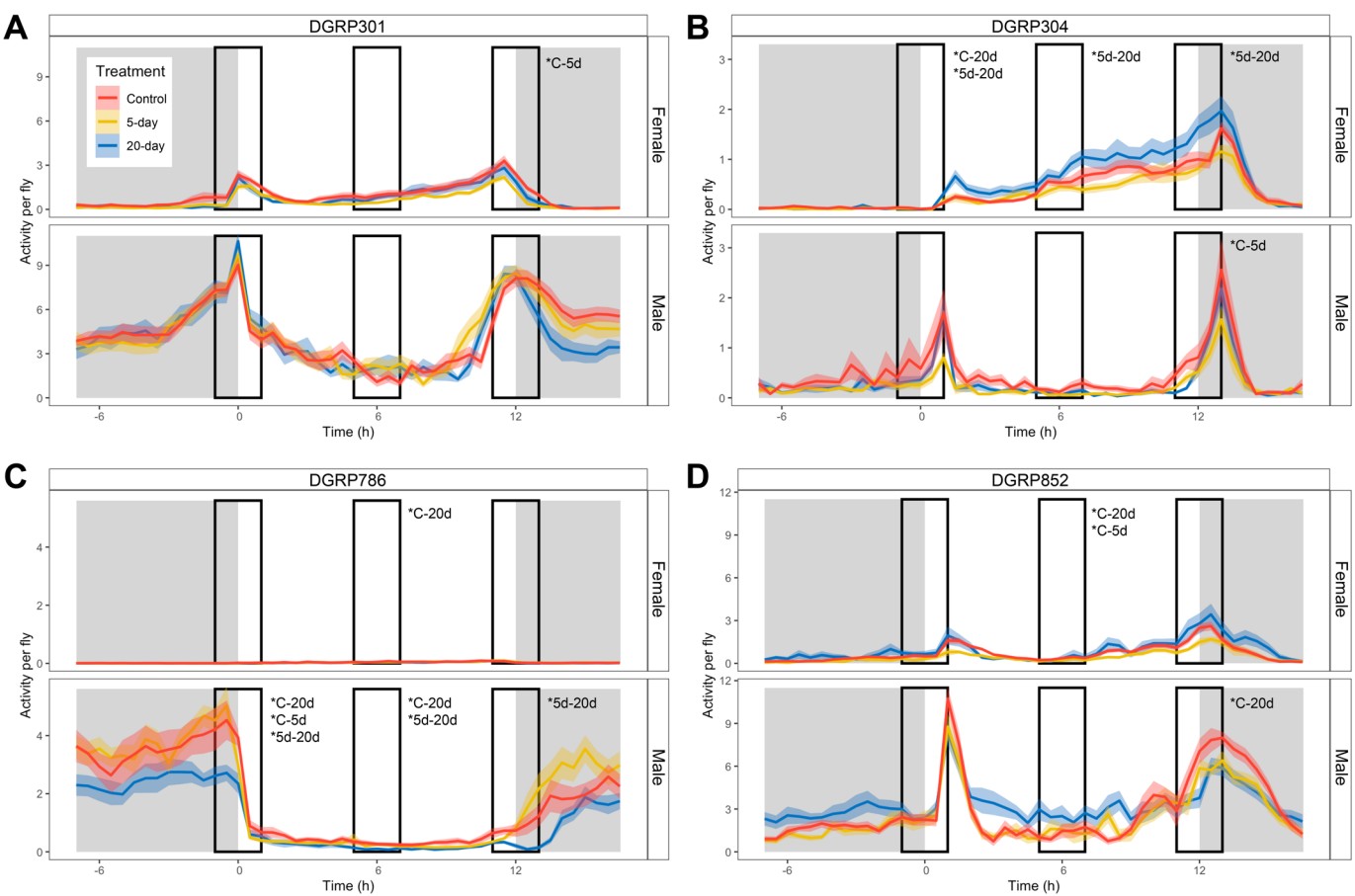

**Fig. 4. The effect of unpredictable disturbance treatment on activity depends on the time of day.** Activity per fly is shown on the y-axis, and time in hours is plotted on the x-axis. Activity is measured using the DAMs, which record the number of times flies cross the beam at the midsection of the vial every 5 min. Results from females (top panel) and males (bottom panel) are displayed from animals 26 days old. Data from the control group (red), 5-day treatment (yellow), and 20-day treatment (blue) show the effect of disturbance on activity per genotype – DGRP301 (A), DGRP304 (B), DGRP786 (C), and DGRP852 (D). The solid-colored lines represent average activity, and the light shading represents s.e. The grey shaded areas indicate lights off, and the white areas indicate lights on, with time 0 marking lights on at 07:00. The black boxes show activity levels from 06:00-08:00 (left), 12:00-14:00 (middle), and 18:00-20:00 (right). * indicates significant differences ($P<0.05$, pairwise $t$-test). Note that the y-axis is not scaled across genotypes. $n$=29-31 vials/group (A), $n$=27-32 vials/group (B), $n$=31-32 vials/group (C), and $n$=23-31 vials/group (D).

compared to controls ($P$=0.023, pairwise $t$-test). In contrast, the 20-day treatment group of the DGRP852 females exhibited higher activity levels from 12:00-14:00 compared to controls ($P$=0.0049, pairwise $t$-test). The only treatment effect detected among DGRP852 males was from 18:00-20:00, with the 20-day treatment group showing a 28% reduction in mean activity compared to the controls ($P$=0.013, pairwise $t$-test; Fig. 4D). These examples illustrate how unpredictable disturbance treatments can show an effect on activity only at a specific time period, as opposed to the entire day. Together, these data demonstrate that if disturbance treatment alters animal activity levels, they will either (1) have the same increased or decreased effect throughout the whole day or (2) have a very specific effect on time periods of high or low natural activity.

### Unpredictable disturbance during early life has little to no effect on lifespan

In our final analysis, we determined how unpredictable disturbance impacts lifespan. Longevity assays were completed for each genotype and included the 5-day and 20-day disturbance treatment groups and a shared control group (Fig. 5). As reported by others, we found that sex ($P$=1.94e-13, Kruskal–Wallis rank sum test) and genotype ($P$<2.2e-16, Kruskal–Wallis rank sum test)

strongly influence lifespan (Fig. 5). When investigating whether unpredictable disturbance treatments alter lifespan, we found significant treatment effects on the median lifespan for DGRP301 females, DGRP304 males, and DGRP852 males (Fig. 5). DGRP301 females showed decreased longevity for the 20-day treatment group, with a decrease in median lifespan from 58 to 56 days ($P$=0.0012, pairwise $t$-test). DGRP852 males also showed a decrease in median lifespan from 53 to 51 days after experiencing a 5-day or 20-day disturbance treatment ($P$=0.011; $P$=0.048, pairwise $t$-test). For the DGRP304 males, the 5-day and 20-day disturbance treatments led to a 10% median lifespan extension, with the median lifespan increasing from 59 to 67 for the 5-day treatment group ($P$=2.3e-06, pairwise $t$-test) and from 59 to 64 for the 20-day treatment group ($P$=8.2e-07, pairwise $t$-test). Overall, these results indicated that unpredictable disturbance can impact lifespan in *Drosophila* in a genotype and sex-specific manner. Though, for most groups, the 5-day and 20-day unpredictable disturbance treatments administered had modest effects on lifespan.

### DISCUSSION

Our results demonstrate that unpredictable disturbance can have a measurable effect on *D. melanogaster* movement behaviors and

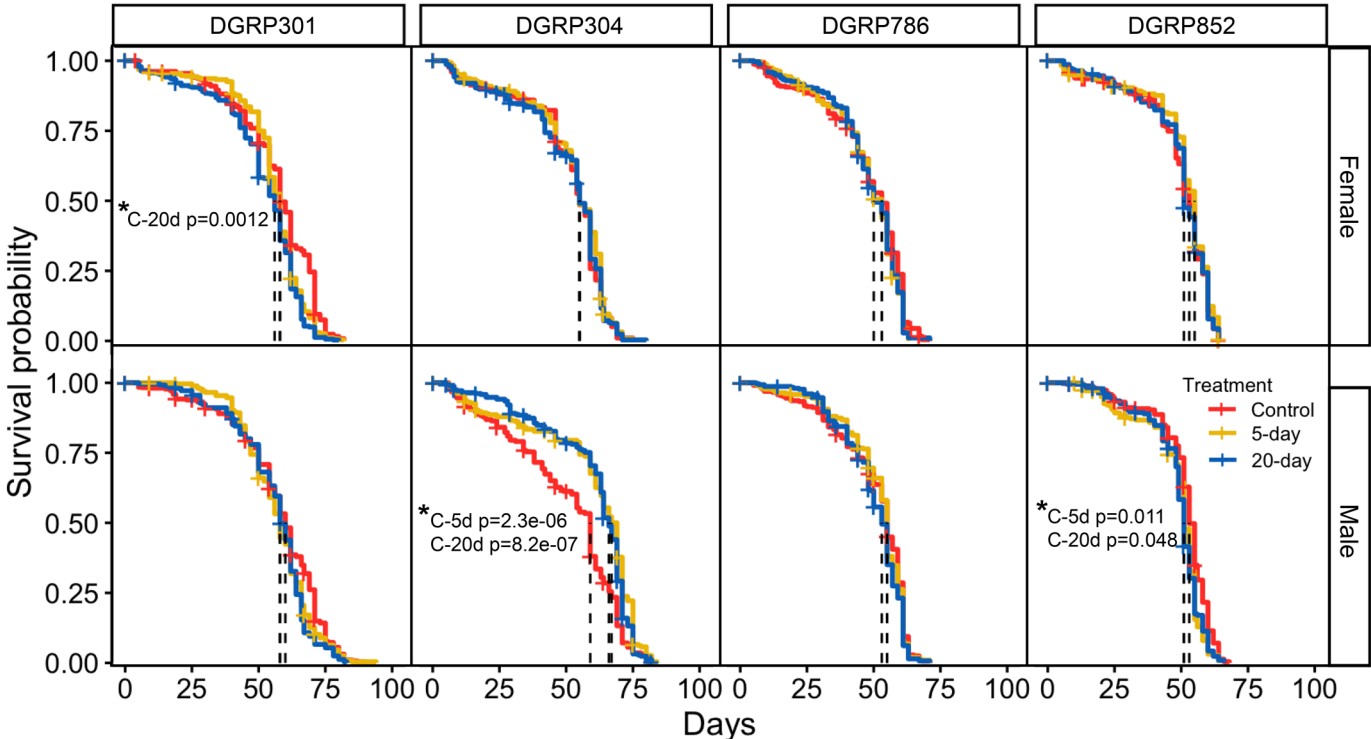

**Fig. 5. Unpredictable disturbance treatments impact lifespan in a genotype and sex dependent manner.** Survival probability (y-axis) is plotted against days (x-axis). Data are shown for females (top panel) and males (bottom panel). Data from the control (red), 5-day treatment (yellow), and 20-day treatment (blue) show that unpredictable disturbance treatments impact lifespan in a genotype and sex dependent manner. The genotypes, DGRP301 (*n*=31 vials), DGRP304 (*n*=27 vials), DGRP786 (*n*=32 vials), and DGRP852 (*n*=31 vials), are labeled across the top. The dashed lines mark the median lifespan of the three groups compared, and significant *P*-values (*t*-tests) are shown.

lifespan, but the effect is context dependent. The four DGRP strains we selected varied in baseline activity levels based on sex and genotype. After modeling human disturbance with an unpredictable disturbance treatment, we detected several significant disturbance-induced changes to total animal activity. Outcomes varied immensely based on the unique sex/genotype/age combination and time of day. Comparing movement at 06:00-08:00, 12:00-14:00, and 18:00-20:00, we noticed that unpredictable disturbance does not always have a detectable effect at all three time periods, and if it does have an effect, it will alter activity in the same direction – either increased or decreased. Finally, early-life disturbance has little to no effect on lifespan, with the exceptions of DGRP301 females, DGRP304 males, and DGRP852 males. Our data illustrate that predicting individual responses to unpredictable disturbance is complex due to sex, genotype, and age all contributing to the outcome.

The data presented here for unpredictable disturbance are consistent with our earlier studies using predictable disturbance treatments to mimic exercise. Similar to what was seen here, these studies consistently showed differences in the baseline activity of different DGRP strains based on sex and genotype, as well as their interaction (Watanabe et al., 2020; Watanabe and Riddle, 2017). Inducing exercise with the TreadWheel also led to variation in outcome measures such as climbing performance and body composition that was dependent on the interaction between sex and genotype (Mendez et al., 2016; Riddle, 2020; Watanabe and Riddle, 2021b). This study extends the findings of an earlier study that used single, 2-h treatments on the TreadWheel for 5 or 20 days to study early-life exercise, which found that sex, genotype, and age strongly influence how animal activity changes in response to these

treatments (Johnson and Riddle, 2024). Despite changing the TreadWheel treatments from a predictable 2-h time interval to four unpredictable 30-min time intervals, sex, genotype, and age impact the phenotypes through interaction effects in both studies (Johnson and Riddle, 2024). Furthermore, in both studies, early-life treatments on the TreadWheel had little to no effect on lifespan, with a few exceptions (Johnson and Riddle, 2024). The results presented here and from our previous work suggest that the impact of TreadWheel treatments typically differs considerably across experimental groups, with effects depending on the sex, genotype, and age of the disturbed animals.

Other studies in *D. melanogaster* also report sex and genotype differences in activity levels following treatments disrupting normal animal activity patterns (Bazzell et al., 2013; Koh et al., 2006; Piazza et al., 2009; Rakshit et al., 2013; Safdar and Wessells, 2023; Sujkowski et al., 2017; Thompson et al., 2020; Vaccaro et al., 2016). For example, Rakshit and colleagues administered a 1 h/day exercise treatment for 5 days/week over 2 weeks and found that exercise lowered average daily activity for wild-type Canton S females but had no effect on males (Rakshit et al., 2013). Another study investigated endurance exercise treatments in two wild-type strains, Canton S and Berlin K, and reported that genotype significantly determined how long flies could keep running before fatiguing (Bazzell et al., 2013). In addition to supporting the importance of sex and genotype factors, several studies also found that older animals respond differently to disturbance than younger animals and that the effect of disturbance can differ based on the time of day (Boomgarden et al., 2019; Krittika and Yadav, 2022; Melnattur et al., 2020; Piazza et al., 2009; Rakshit et al., 2013; Zheng et al., 2015). Zheng and colleagues tested several exercise

regimens in wild-type females and observed a treatment effect on climbing height of 6-week-old flies but not 2-week-old flies (Zheng et al., 2015). They also detected differences in daily activity between control and exercised 6-week-old flies but no differences in evening activity between these groups (Zheng et al., 2015). Krittika and Yadav noticed that male wild-type flies responded differently to monochromatic light disturbance, with only blue and green light conditions causing a significantly higher ramp-up of activity in anticipation of the lights turning off (Krittika and Yadav, 2022). Thus, the available data suggest that the effect of disturbance is variable with age and time of day, with animals able to respond differently at various time points. Together, these results confirm that predicting the effect of disturbance is complex and requires understanding how sex and genotype interact to influence responses.

Reports from other species share many of the findings from the *D. melanogaster* studies. Multiple groups, utilizing rodents, fish, and worms, found that sex and genotype influence the response to various treatments disturbing natural activity patterns in these species (Adamovich et al., 2021; Banks et al., 2022; Earnest et al., 2016; Hiramatsu and Garland, 2018; Kelly et al., 2014; Sullens et al., 2025; Wierczeiko et al., 2021). Hiramatsu and Garland selectively bred mice for higher wheel-running and reported sex-by-genotype interactions for four mouse strains in terms of voluntary running behavior (Hiramatsu and Garland, 2018). Banks and colleagues simulated shift work with two different light/dark cycle disruptions in wild-type mice and discovered that female and male mice exhibited different wheel running behavior after these disturbances (Banks et al., 2022). Both sexes showed lower voluntary wheel running after exposure to an alternating light/dark cycle schedule, but the reduction of activity was more severe in females (Banks et al., 2022). In other models, age and time of day also influence responses to treatments disturbing activity patterns (Adamovich et al., 2021; Berio et al., 2023; Guidi et al., 2023; Hsieh et al., 2014; Saderi et al., 2019; Salgado-Delgado et al., 2013). Berio and colleagues investigated swimming behavior in 28°C versus 32°C conditions of larval, juvenile, and adult stage zebrafish (Berio et al., 2023). The warmer temperature increased tail beat frequency in the larvae stage but not in the juvenile or adult stages, indicating that the fish were more sensitive to temperature disturbance earlier in life (Berio et al., 2023). Guidi and colleagues investigated changes in zebrafish activity in response to mercury and arsenic pollutants (Guidi et al., 2023). Measurements of locomotor activity after mercury exposure revealed a time-of-day dependent disturbance effect, as researchers saw more of a reduction in swimming at mid-light compared to mid-night (Guidi et al., 2023). Another example of time-dependent effects was uncovered by Hsieh and colleagues when wild-type rats were forced to be active during normal periods of rest to simulate shift work (Hsieh et al., 2014). After 4-5 weeks of this treatment, they detected the most significant reductions in activity at lights on and lights off (Hsieh et al., 2014). These examples illustrate that the influence of sex, genotype, age, and time of day in treatments disturbing natural activity patterns is not limited to *D. melanogaster*. In addition, this work from other model organisms supports the conclusion that disturbance can have a range of effects on activity that are difficult to predict.

Other studies also confirm that treatments disrupting natural activity patterns often have minimal impacts on lifespan. In agreement with our findings, many *Drosophila* studies found significant contributions of sex and genotype to lifespan (Ebanks et al., 2021; Klarsfeld and Rouyer, 1998; Lateef et al., 2023; Le Bourg, 2021; Le Bourg et al., 2001; Le Bourg and Polesello, 2019; Shen et al., 2019; Sujkowski et al., 2015, 2012; Thompson et al., 2020; Vaccaro et al., 2016). In addition, studies conducted in healthy, wild-type flies often detected little to no effect of disturbance on lifespan (Ebanks et al., 2021; Lateef et al., 2023; Le Bourg, 2021; Le Bourg et al., 2001; Le Bourg and Polesello, 2019; Rakshit et al., 2013; Sujkowski et al., 2015). Likely, the disturbance treatments used in these studies, like the 30 min disturbance treatment used here, represent a relatively minor disruption of natural behaviors. In contrast, impacts on lifespan often are seen in studies using sick animals or chronic disturbance treatments. For example, a brief mechanical sleep disruption of 8 min/day over 2 days did not alter the survival of wild-type *D. melanogaster,* but flies with a *timeless* gene mutation did exhibit reduced lifespan (Lateef et al., 2023). $Tim^{01}$ mutants were arrhythmic, meaning their natural circadian behavioral rhythms were completely eliminated (Lateef et al., 2023). Short-term exercise treatments administered to healthy, wild-type flies had small to no effect on lifespan (Ebanks et al., 2021; Rakshit et al., 2013; Sujkowski et al., 2015), but when Wen and colleagues fed *D. melanogaster* different high-salt diets, they found that disturbance-inducing exercise significantly extended the lifespan of some animals (Wen et al., 2020). In addition, several studies report that altering animal movement patterns through chronic light/dark cycle disruption impacted the survival of both wild-type and mutant *D. melanogaster* lines (Boomgarden et al., 2019; Shen et al., 2019; Vaccaro et al., 2016). For example, a chronic circadian misalignment treatment involving a 4-h delay in the light/dark cycle, reduced median lifespan in male and female wild-type animals (Boomgarden et al., 2019). Taken together, the available data suggest that short-term treatments disturbing normal activity patterns in *D. melanogaster* typically do not alter the lifespan of healthy animals, particularly if the treatments are limited to young ages. In contrast, disturbance treatments for animals of poor health with suboptimal diet or sleep, or long-term treatments appear to negatively affect survival.

Similar results regarding the impact of treatments altering natural activity patterns on lifespan are seen in other species as well. Like in *D. melanogaster*, sex and genotype influenced lifespan in several rodent studies (Earnest et al., 2016; Hazell et al., 2024; Swindell, 2012; Toth et al., 2017), and a *C. elegans* study recorded lifespan differences linked to stress resistance in long-lived mutants (Soo et al., 2022). As reported here, disturbance treatments had little to no effect on healthy, wild-type mice (Earnest et al., 2016; Garcia-Valles et al., 2013; Samorajski et al., 1985; Swindell, 2012), while treatments disrupting natural circadian rhythms showed impacts on lifespan in animals suffering from poor health or experiencing chronic levels of disruption (Ramsey et al., 2023; Toth et al., 2017). For example, chronically exposing cancer-prone female mice to an alternating light/dark cycle increased their death rate (Toth et al., 2017). Similarly, chronic circadian disruption with a constantly advancing light cycle in stroke-prone, hypertensive rats resulted in a major decrease in percent survival compared to non-disturbed controls (Ramsey et al., 2023). Thus, data from other species support our finding that short-term, unpredictable disturbance of normal activity patterns does not modify lifespan in healthy animals and reflects a broader pattern seen across diverse animal species.

Our findings and the work of others in diverse species demonstrate that context is important in understanding whether disturbance will be beneficial, harmful, or have no obvious impact on an animal's health. Extending these results from laboratory studies like ours to natural populations suggests that the type of

human disturbance and the animal's unique genetic makeup will influence whether the disturbance is harmful and if so, to what degree. Our work suggests that in the wild, animals are likely to respond differently to human disturbance based on their sex, age, and natural baseline activity levels. However, it also illustrates that the long-term impacts of short-term disturbance events are relatively minor and that there are genotypes that are resistant and do not show detrimental effects. Generally, data available suggest that healthy animals, not experiencing other stressors, might be relatively resilient to some human disturbance (Holzner et al., 2021; Thurfjell et al., 2017), as illustrated by our finding that healthy, wild-type flies showed little alteration in their lifespan following disturbance. In contrast, these data suggest that natural populations or individuals experiencing additional stressors, such as pollution, limited access to food, or habitat loss, might be more likely to exhibit negative side effects of disturbance, such as decreased survival (Kerbiriou et al., 2009; López-García et al., 2021; Schuyler et al., 2023). For those animals that are impacted by disturbance, it might be possible to alleviate the effects by reducing additional stressors, e.g. through restoration of their habitat (Elo et al., 2015; Hardison et al., 2023; Piczak et al., 2024) or food supplementation. In summary, our results highlight the importance of considering sex, genotype, age, and time of day factors when predicting the effects of unpredictable disturbance on activity and lifespan.

Given the widespread impacts of human disturbance, it is important to test additional conditions in the laboratory, especially if the reduction of other stressors can alleviate the detrimental effects of disturbance. As a follow-up to this study, several different avenues can be envisioned. For example, it would be of interest to investigate a wider range of disturbance treatments, either varying the length of the treatment or timing across the fly's lifespan, changing the intensity of the daily disturbance, or combining the disturbance with an additional challenge that an animal might experience, such as a decreased food quality. These types of studies would improve our understanding of the range of responses that disturbance regimens can elicit in *D. melanogaster* and under what conditions. Also, of interest would be to discover the genetic factors that impact the response of *D. melanogaster* to this type of disturbance. This information could be discovered by expanding this study to include additional DGRP lines to make it possible to carry out a genome-wide association study (GWAS) to identify genetic loci linked to variation in response to the disturbance. While our study documents that disturbance responses are sex- and genotype-specific and presents compelling evidence that *D. melanogaster* can be utilized to gain insights into disturbance, additional studies are needed to gain a comprehensive understanding of the factors controlling responses to disturbance.

## MATERIALS AND METHODS
### *Drosophila* stocks and husbandry
All fly stocks used in this study belong to the DGRP and were obtained from the Bloomington *Drosophila* Stock Center (Bloomington, IN, USA) (Huang et al., 2014; Mackay et al., 2012). We used DGRP301, DGRP304, DGRP786, and DGRP852 (RRIDs: BDSC_25175, BDSC_25177, BDSC_25206, BDSC_25209). Flies were housed in temperature-, humidity-, and light-controlled incubators at 25°C, 60% humidity, and a 12 h:12 h light/dark cycle. All flies were reared on Fisherbrand Jazz-Mix food (Waltham, MA, USA) in bottles supplemented with active yeast, containing 20 females and 10 males. Newly eclosed flies were anesthetized using $CO_2$, collected as virgins in 8-h time windows, separated by sex, and aged for 5 days to be used in the experiments. After collection, flies were housed in same sex groups of 20 animals per vial with 1-inch of food and yeast paste. These collection procedures were repeated on

four separate occasions for each of the four genotype cohorts. Flies were transferred onto new food every 3-4 days. At 5 days old, flies that were assigned to the 5-day or 20-day group were loaded onto the TreadWheel to begin the unpredictable disturbance treatment (Fig. 1A). The animals assigned to the control group were placed next to the TreadWheel. Throughout the treatment, all flies were maintained in the incubator under standard conditions.

### Unpredictable disturbance treatment
Disturbance treatments were administered using the TreadWheel. As described by Mendez and colleagues, the TreadWheel induces movement by rotating the animal enclosures and exploiting the animals' natural tendency to reach the top of their enclosure, known as negative geotaxis (Mendez et al., 2016). The motor was calibrated to four rotations per minute, a speed which reliably elicits movement in these animals, while minimizing stress (Mendez et al., 2016). Responding to their natural negative geotaxis, the animals climb the vial walls but can also choose to rest. The TreadWheel was placed in an incubator under standard conditions, 25°C, 60% humidity, and a 12 h:12 h light/dark cycle. The flies were disturbed at four random 30-min intervals per day (2 h total/day) over either 5 or 20 days, while the controls were housed alongside treated flies but received no disturbance treatment. These 30-min treatment periods could occur at any time between 12:00-23:59 and were randomly set from day to day, with each cohort receiving treatment at different random times. Fifteen vials were used for each experimental group (sex/genotype/treatment) (Fig. 1A). Once loaded onto the TreadWheel, the animals remained there until the last day of disturbance treatment. The last day of disturbance for the 5-day group was on day 9, and the last day for the 20-day group was on day 24. For each experimental group, flies were collected at ages 0-1 days old and disturbed from 5-9 days old (5-day treatment) or 5-24 days old (20-day treatment).

### Fly activity recordings
Before and during activity recordings, flies were maintained in incubators under standard husbandry conditions, as previously described. Baseline activity levels were determined using the TriKinetics *Drosophila* Activity Monitoring (DAM) system (Waltham, MA, USA), following a protocol similar to that described by Woods et al. (2014). DAM recordings were taken over a 3-day period. On the first day, flies were transferred onto new food before vials were placed vertically in the DAMs, and flies were allowed to adapt. On the second day, activity was monitored continuously in 5-min intervals over 24 h in incubators. Finally, vials were removed from the DAMs on the third day. We only used activity data from the second day, when vials were on the DAMs for a full 24 h and any potential effects of handling were eliminated. Activity data was taken at four ages: days 11, 26, 37, and 51 (Fig. 1B). Specifically, the activity of the controls and 5-day treatment group was recorded on day 11, immediately following the 5-day disturbance treatment. Activity of all three groups was recorded on day 26 immediately following the 20-day disturbance treatment and at later ages on days 37 and 51. Activity data were normalized by fly number, resulting in a measurement of activity/fly.

### Lifespan assay
Newly emerged flies were collected over 1-2 days and housed separately by sex in vials of 20. A total of 900 male flies and 900 female flies were collected for each genotype. These animals were randomly assigned to the control, 5-day, or 20-day groups. All flies were kept in incubators under standard husbandry conditions, as described earlier. To record lifespan, deaths were recorded daily during the disturbance treatment and every other day after the treatment ended (Fig. 1B). Counts started at the age of day 5 and continued until the last fly in each group died. Deaths were not recorded at the same time every day. During the assay, the flies were transferred onto new food every 3-4 days without $CO_2$ (Fig. 1B). The flies used in the lifespan assay were the same animals used for the fly activity recordings.

### Data analysis
Statistical analyses were performed using RStudio version 2024.09.0+375 (2024.9.0.375) and R version 4.4.1 (2024-06-14) (Posit team, 2024).

Boxplots of daily activity were created using the package *ggplot2* version 3.5.1. Line graphs of activity vs time were prepared using *ggetho* version 0.3.7 and an adaptation of the *Rethomics* framework (Geissmann, 2022; Geissmann et al., 2019). Shapiro tests, histograms, and qq-plots were used to assess normality (Posit team, 2024). As most of the data were not normally distributed, we used a linear mixed effects type 3 ANOVA with the lme4 R package (Bates et al., 2015) and a repeated measures ANOVA. Pairwise comparisons were made using pairwise *t*-tests. We used the *stats* package version 4.4.1 to perform the ANOVAs and *t*-tests (Posit team, 2024). Outliers, defined by the interquantile range method, were removed prior to the final analysis, as this step improved normality and model fit (models based on data with and without outlier removal were compared using Akaike information criterion; for details see the analysis scripts provided on the Riddle lab GithHub page). Lifespan was analyzed with the *survival* package version 3.8-3 (Therneau, 2024; Therneau and Grambsch, 2000). Survival curves were plotted using a Kaplan–Meier estimate. Differences between lifespan curves were evaluated using a log-rank test (*survdiff*), while differences in mean survival were evaluated using ANOVA and *t*-tests. A false discovery rate was used to adjust for multiple testing. For analysis details, see the data and analysis scripts that are available on the Riddle lab GitHub page (https://github.com/riddlenc/Byars_2025.git).

## Acknowledgements
We thank the Riddle lab members for their helpful discussions and feedback. Special appreciation goes to Heidi Johnson for her guidance as A.S.X.B.'s bench mentor during the initial phases of this project. A.S.X.B. was supported by an Alabama Space Grant Consortium (ASGC) scholarship. Stocks obtained from the Bloomington *Drosophila* Stock Center (NIH P40OD018537) were used in this study.

## Competing interests
The authors declare no competing or financial interests.

## Author contributions
Conceptualization: A.S.X.B., N.C.R.; Data curation: A.S.X.B., N.C.R.; Formal analysis: A.S.X.B., N.C.R.; Methodology: N.C.R.; Project administration: N.C.R.; Resources: N.C.R.; Supervision: N.C.R.; Visualization: A.S.X.B., N.C.R.; Writing – original draft: A.S.X.B., N.C.R.; Writing – review & editing: A.S.X.B., N.C.R.

## Funding
 Deposited in PMC for immediate release.

## Data and resource availability
Our data is publicly available on the Riddle Lab GitHub page under the repository name Byars_2025 (https://github.com/riddlenc/Byars_2025.git) and in the supplementary information.

## First Person
This article has a related First Person interview with the first author of the paper.

## Peer review history
The peer review history is available online at https://journals.biologists.com/bio/lookup/doi/10.1242/bio.062071.reviewer-comments.pdf

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
