## [Peer Review File · Biology Open]

Unpredictable disturbance and its effects on activity behavior and lifespan in *Drosophila melanogaster*

Anabel S. X. Byars and Nicole C. Riddle

DOI: 10.1242/bio.062071

Editor: Kendra J. Greenlee

Review timeline

Original submission:	15 May 2025
Editorial decision:	22 May 2025
First revision received:	13 June 2025
Accepted:	17 June 2025

Original submission

First decision letter

MS ID#: bio.062071

MS TITLE: Unpredictable disturbance and its effects on activity behavior and lifespan in *Drosophila melanogaster*

AUTHORS: Anabel S. X. Byars; Nicole C. Riddle

I have now reached a decision on the above manuscript.

The reviewer reports are shown at the bottom of this email or can be accessed, together with a copy of this decision letter, by going to:

As you will see, the reviewers raised a number of substantial criticisms that prevent me from accepting the paper at this stage.

They suggest, however, that a revised version might prove acceptable, if you can address their concerns. If you think that you can deal satisfactorily with the criticisms on revision, I would be pleased to see a revised manuscript. We would then return it to the reviewers.

At this stage, we also ask you to ensure your manuscript complies with our formatting guidelines. Provided you are able to fully address the referees' comments, we are positive about publication of your paper (we accept over 95% of revision submissions) and therefore hope you won't mind any extra work involved in reformatting your manuscript at this point.

Please ensure that you clearly highlight all changes made in the revised manuscript. Please avoid using 'Tracked changes' in Word files as these are lost in PDF conversion.

I should be grateful if you would also provide a point-by-point response detailing how you have dealt with the points raised by the reviewers in the 'Response to Reviewers' box. Please attend to all of the reviewers' comments. If you do not agree with any of their criticisms or suggestions please explain clearly why this is so.

Reviewer 1

Comments for the author

This manuscript examines the impact of unpredictable environmental stimuli on longevity in genetically diverse populations of flies. The authors use a TreadWheel to disturb the flies in an unpredictable fashion and measure the effects on sleep and longevity. A strength of the manuscript is the use of genetically variable DGRP lines, and an ethologically relevant behavioral paradigm. The manuscript is clearly written and the experiments are statistically sound. Overall, additional characterization of DGRP lines will be useful to the community. It is unfortunate that the major hypothesis yielded negative results, and the manuscript would benefit from additional description of how experiments might be modified to examine the role of unpredictable disturbances and longevity. The points below are meant to provide guidance for how the manuscript could be improved. In particular, please address points #7, 8, 9, and 11.

-

1. The focus of the introduction is not entirely clear, and seems to stray from the primary area of study. Directing the introduction towards unpredictable stimuli and stress would increase accessibility.
2. It would be help to discuss flies as models for studying sleep deprivation and stress, which presumably are impacted by unpredictable disturbances.
3. Line 154: Specify the age of flies at collection and whether they were housed separately;
4. Is TreadWheel stress, forced activity, or disturbance? More detail on what the perceived manipulation does to the fly would be helpful.
5. Line 185: How is the effect of the disturbance on activity a measure of health?
6. Line 189: Why were these four DGRP lines chosen? There is significant sleep and activity diversity across different DGRP lines.
7. Female 786 look like they are not moving at all. Please check data, and perhaps zooming in would be helpful.
8. Inactivity for five minute periods is typically defined as sleep. This could be an additional analysis that broadens the scope of the manuscript.
9. Line 211: The description of statistics centers around interactions. Please state at which ages was there a significant effect compared to undisturbed controls.
10. Line 431. A major part of this is the degree to which natural behaviors are impacted. It is important to expand on variables that might be changed to elicit an effect on lifespan.
11. Line 441. There is evidence that sleep deprivation reduces lifespan in flies including chronic sleep deprivation experiments, and that many sleep mutants have shorter lifespans.
12. A major application of the DGRP collection is the ability to map genetic variation onto behavior. It would be useful to highlight this and discuss how scaling up the number of lines tested could be used to map genes.

Reviewer 2

Comments for the author

In this research article, the authors use a TreadWheel device to administer randomized disturbances to four different *Drosophila melanogaster* strains - DGRP301, DGRP304, DGRP786, and DGRP852. They assess how these unpredictable disturbances, mimicking human disturbances, impact the locomotor activity and lifespan of the flies. The findings demonstrate that individual fly responses to unpredictable disturbances is complex and context-dependent, influenced by the sex, genotype, and age of the flies. These observations are consistent with the previous predictable disturbance studies that mimic regular exercise using the TreadWheel behavioral prototype developed by the same group.

While the experimental approach supports the study's conclusion, the manuscripts could benefit from improvements in clarity, presentation, and reproducibility. Below are some suggestions for the authors to consider:

1. Include the genus and species names, "*Drosophila melanogaster*," in the title.

2. Revise the keyword list to exclude terms already present in the title, such as "unpredictable disturbance" and "lifespan". Suggested alternatives include "locomotor activity," "circadian rhythms," "TreadWheel,"
3. Line 34-35: Consider including the search for a mating partner and escape from threats/predators.
4. Line 63: clarify the effects reported by Doherty and colleagues by specifying the reported effects for clarity.
5. Lines 91-94 are too vague and lack citations; consider revising them for clarity.
6. For reproducibility, consider providing sufficient detail about the experimental method or citing previously published protocols for additional information in the method section. For example,
 - a. Lines 125-126, clarify how the experimental flies are maintained after collection, specifically, transfer schedule for the 20-day assay.
 - b. Line 138: "Fifteen vials were used for each experimental group (sex/genotype/treatment)." It is unclear whether all replicates were tested on the same day or across different days, whether the assay was conducted at the same time of the day across all tested conditions, whether all four strains were tested at the same time.
 - c. For fly activity recordings, it is recommended to cite previously published protocols, as the current description is brief.
 - d. Lines 147-149, include details on how flies were handled and maintained before and after recordings.
 - e. Line 154, "To record lifespan, deaths were recorded daily during the disturbance treatment and every other day after the treatment ended." Specify the time of day when lifespan data were collected.
 - f. Lines 156-158, consider providing a schematic of the experimental workflow for clarity.
 - g. Throughout the manuscript, indicate the statistical tests used to determine significance values adhering to the journal's formatting instead of generically mentioning t-test or ANOVA.
7. Line 178-179, include the corresponding reference. More broadly, ensure that citations are consistently provided throughout the manuscript.
8. In various sections of the manuscript, mention the figure or table number appropriately. For instance, Lines 243-250, 258-263, and 329-331.
9. On Lines 307-309, When referring to DGRP786 females, note the absence of a corresponding activity graph in Figure 3C. Based on Figures 2 and S1, this may refer to male activity data. Please verify and revise to correct any inconsistencies.
10. Figure 2 is essentially a miniature version of Figure S1, highlighting significant differences in activity observed in three strains at different time points and sex. Rather than duplicating, consider adjusting the Y-axis scale to effectively visualize the effects.
11. For clarity and readability:
 - a. Present control data (grey in box plots) first, followed by data for 5- and 20-day conditions.
 - b. Consider changing the color schematics in the activity plots in Figures 3 and 4, as the current grey background makes it difficult to visualize the control group.
 - c. Ensure the Y-axis units are clearly labeled across in graphs.
12. A brief section mentioning the study's limitation could benefit the discussion. Of curiosity, have the authors tested any harsh disturbance protocols - increased number of rotations per minute, more than 2-hr disturbances? Can flies habituate to these predictable and/or unpredictable disturbances? Has other tested other *Drosophila* species for these disturbance protocol?

Reviewer's Responses to Questions

Experimental quality

Does each figure have the proper controls?

If 'No', please indicate reasons in Comments for Author box below.

Reviewer #1:

- Yes

Reviewer #2:

- Yes
-
-

Were the data analyzed using appropriate statistical tests?

If 'No', please indicate reasons in Comments for Author box below.

Reviewer #1:

- Yes

Reviewer #2:

- No
-
-

Reproducibility

Were experiments performed using adequate number of biological replicates?

If 'No', please indicate reasons in Comments for Author box below.

Reviewer #1:

- Yes

Reviewer #2:

- No
-
-

Does the methods section provide sufficient detail to permit reproducibility?

If 'No', please indicate reasons in Comments for Author box below.

Reviewer #1:

- Yes

Reviewer #2:

- No

Completeness

Are the manuscript's conclusions supported by the data?

If 'No', please indicate reasons in Comments for Author box below.

Reviewer #1:

- No

Reviewer #2:

- Yes
-
-

Scholarship

Do the authors cite and discuss the merits of data that would argue for and against their conclusion?

If 'No', please indicate reasons in Comments for Author box below.

Reviewer #1:

- Yes

Reviewer #2:

- Yes
-
-

Does the manuscript title & abstract accurately reflect the contents of the manuscript, without hyperbole?

If 'No', please indicate reasons in Comments for Author box below.

Reviewer #1:

- Yes

Reviewer #2:

- Yes
-

First revision

Author response to reviewers' comments

Dear Dr. Greenlee,

We appreciate the detailed comments provided by the two reviewers and the opportunity to revise our manuscript in response to their comments. Below, is a point-by-point response to the reviewers' comments, indicating how we have used the suggestions to improve our manuscripts. Any changes we have made are highlighted in the Word document of the manuscript we submitted for this revision. We trust that you find our manuscript much improved and now suitable for publication in *Biology Open*.

Best,

Nicole C. Riddle on behalf of the authors

Response to Reviewer 1

The focus of the introduction is not entirely clear and seems to stray from the primary area of study. Directing the introduction towards unpredictable stimuli and stress would increase accessibility.

We have edited throughout the introduction to address this point (lines 56-57, 60, 63-64, 74, 92-93, 95-96).

It would be helpful to discuss flies as models for studying sleep deprivation and stress, which presumably are impacted by unpredictable disturbances.

We have added several sentences to the introduction in paragraph four that provide examples using flies to study sleep deprivation and various stressors (lines 102-112).

Line 154: Specify the age of flies at collection and whether they were housed separately.

We have added the requested information in line 187.

Is TreadWheel stress, forced activity, or disturbance? More detail on what the perceived manipulation does to the fly would be helpful.

We have provided more detail in the methods section, explaining the type of disturbance that the TreadWheel applies (lines 156-157).

Line 185: How is the effect of the disturbance on activity a measure of health?

We have rewritten the sentence in question for clarity (lines 222-223).

Line 189: Why were these four DGRP lines chosen? There is significant sleep and activity diversity across different DGRP lines.

We have added a sentence explaining why these particular DGRP lines were selected (lines 228-230).

Female 786 look like they are not moving at all. Please check data, and perhaps zooming in would be helpful.

We agree with the reviewer that the data for DGRP786 females was difficult to examine due to the y-axis scaling and the large difference in activity between females and males. To address this issue, we created a new Supplemental Figure 1 that display DGRP786 female activity only.

Inactivity for a five-minute periods is typically defined as sleep. This could be an additional analysis that broadens the scope of the manuscript.

For sleep studies, flies are typically housed individually. We housed 20 flies per vial, making it difficult to assess sleep, unless all animals were asleep. Given this limitation, we decided to not

include an analysis focused on sleep. Based on the reviewers suggestion, we plan to determine if data from cohorts of flies can provide insights into sleep for future work.

Line 211: The description of statistics centers around interactions. Please state at which ages was there a significant effect compared to undisturbed controls.

We have added a sentence stating at which ages there were significant disturbance treatment effects compared to undisturbed controls (lines 257-260).

Line 431. A major part of this is the degree to which natural behaviors are impacted. It is important to expand on variables that might be changed to elicit an effect on lifespan.

We agree with the reviewer and have reworded and expanded paragraph 5 of the discussion section (lines 480-485) to demonstrate how disrupting natural behaviors like diet and sleep produces sickly flies that are more vulnerable to disturbance effects on lifespan.

Line 441. There is evidence that sleep deprivation reduces lifespan in flies including chronic sleep deprivation experiments, and that many sleep mutants have shorter lifespans.

We have included additional information about sleep deprivation and sleep mutants in the introduction and discussion (paragraph starting at lines 105 and 486 onward).

A major application of the DGRP collection is the ability to map genetic variation onto behavior. It would be useful to highlight this and discuss how scaling up the number of lines tested could be used to map genes.

We have added text discussing how scaling up the number of DGRP lines and running a GWAS would be a useful next step (lines 548-552).

Response to Reviewer 2

Include the genus and species names, "Drosophila melanogaster," in the title.

We have added *Drosophila melanogaster* to the title.

Revise the keyword list to exclude terms already present in the title, such as "unpredictable disturbance" and "lifespan". Suggested alternatives include "locomotor activity," "circadian rhythms," "TreadWheel,"

We have changed our keywords list based on this helpful suggestion.

Line 34-35: Consider including the search for a mating partner and escape from threats/predators.

We appreciate the reviewer's suggestion and have edited the first paragraph of the introduction (lines 42-45, 48-49, 53-55) to include the escape from predators and an additional example of the search for mating partners.

Line 63: clarify the effects reported by Doherty and colleagues by specifying the reported effects for clarity.

We have clarified the effects reported by Doherty and colleagues by expanding our description of this study (lines 74-76).

Lines 91-94 are too vague and lack citations; consider revising them for clarity.

We have rewritten this section lines for clarity (lines 100-103) and added two citations (lines 100, 103).

For reproducibility, consider providing sufficient detail about the experimental method or citing previously published protocols for additional information in the method section. For example,

- a. Lines 125-126, clarify how the experimental flies are maintained after collection, specifically, transfer schedule for the 20-day assay.*
- b. Line 138: "Fifteen vials were used for each experimental group (sex/genotype/treatment)." It is unclear whether all replicates were tested on the same day or across different days, whether the assay was conducted at the same time of the day across all tested conditions, whether all four strains were tested at the same time.*
- c. For fly activity recordings, it is recommended to cite previously published protocols, as the current description is brief.*
- d. Lines 147-149, include details on how flies were handled and maintained before and after recordings.*
- e. Line 154, "To record lifespan, deaths were recorded daily during the disturbance treatment and every other day after the treatment ended." Specify the time of day when lifespan data were collected.*
- f. Lines 156-158, consider providing a schematic of the experimental workflow for clarity.*
- g. Throughout the manuscript, indicate the statistical tests used to determine significance values adhering to the journal's formatting instead of generically mentioning t-test or ANOVA.*

We added additional information in the methods section. Specifically, we made the following edits:

- a. Lines 142-149: We added the requested information.
- b. Lines 144-145, 161-168: We added the requested information.
- c. Lines 174: We now provide a citation for a JoVE protocol.
- d. Lines 171-184: We added the requested information.
- e. Lines 192-193: We added the requested information.
- f. We created a new Figure 1 showing the experimental design.
- g. Throughout the manuscript, we provided information as to the statistical tests used and added further detail in the methods on lines 203-205.

Line 178-179, include the corresponding reference. More broadly, ensure that citations are consistently provided throughout the manuscript.

We have added a reference to line 216 and proofread the manuscript for any missing references.

In various sections of the manuscript, mention the figure or table number appropriately. For instance, Lines 243-250, 258-263, and 329-331.

We have included additional references to tables and figures in the text (lines 281-289, 294-307, 370-372).

On Lines 307-309, When referring to DGRP786 females, note the absence of a corresponding activity graph in Figure 3C. Based on Figures 2 and S1, this may refer to male activity data. Please verify and revise to correct any inconsistencies.

We do have activity data for DGRP786 females, but the activity is very low compared to the males, which means that the y-axis scaling makes it appear that there are no data. To clarify, we created a new Supplemental Figure 1, depicting the activity of only DGRP786 females with altered y-axis scaling.

Figure 2 is essentially a miniature version of Figure S1, highlighting significant differences in activity observed in three strains at different time points and sex. Rather than duplicating, consider adjusting the Y-axis scale to effectively visualize the effects.

Based on the reviewer's suggestion, we replaced the original Figure 2 with an unscaled version of the old Supplemental Figure 1 to allow the reader to easily compare across all groups. Now, Supplemental Figure 1 focuses on the data from DGRP786 females, with the y-axis scaled to clearly show their activity.

For clarity and readability:

- a. Present control data (grey in box plots) first, followed by data for 5- and 20-day conditions.*
- b. Consider changing the color schematics in the activity plots in Figures 3 and 4, as the current grey background makes it difficult to visualize the control group.*
- c. Ensure the Y-axis units are clearly labeled across in graphs.*

We made changes to address points a, b, and c.

A brief section mentioning the study's limitation could benefit the discussion. Of curiosity, have the authors tested any harsh disturbance protocols - increased number of rotations per minute, more than 2-hr disturbances? Can flies habituate to these predictable and/or unpredictable disturbances? Has other tested other Drosophila species for these disturbance protocol?

We have amended the discussion to address limitations. On lines 540-555, we discuss how the scope of this project could be expanded to examine different intensities of disturbance, timing, types of disturbance, and other species or genotypes. In terms of habituation, we have a small amount of data that suggest that if you treat animals on the Treadwheel, then give them a short break, then re-start the treatment, in the subsequent bouts of disturbance there tends to be less movement (see Figure 4 in <https://doi.org/10.1371/journal.pone.0185090.g004>). It is unclear if this type of habituation would occur with the protocol used here where the time interval between treatments is several hours.

Second decision letter

MS ID#: bio.062071R1

MS TITLE: Unpredictable disturbance and its effects on activity behavior and lifespan in *Drosophila melanogaster*

AUTHORS: Anabel S. X. Byars; Nicole C. Riddle

I am happy to tell you that your manuscript has been accepted for publication in Biology Open, pending our standard publication integrity checks. It was accepted on 17 Jun 2025